# The Use of 3D Printing Technology in Gynaecological Brachytherapy—A Narrative Review

**DOI:** 10.3390/cancers15164165

**Published:** 2023-08-18

**Authors:** Barbara Segedin, Manja Kobav, Helena Barbara Zobec Logar

**Affiliations:** 1Department of Radiation Oncology, Institute of Oncology Ljubljana, 1000 Ljubljana, Slovenia; mkobav@onko-i.si (M.K.); hlogar@onko-i.si (H.B.Z.L.); 2Faculty of Medicine, University of Ljubljana, 1000 Ljubljana, Slovenia

**Keywords:** gynaecological cancer, brachytherapy, applicator development, 3D printing, additive manufacturing

## Abstract

**Simple Summary:**

Cervical and endometrial cancers are the fourth and sixth most common cancers in women, respectively. Radiation therapy, including brachytherapy, is an important component of their treatment. Commercially available brachytherapy applicators only come in limited sizes and designs and either do not fit in some patients or do not allow adequate dose delivery to the target volume. In recent years, customised 3D-printed applicators have been increasingly used in such cases. This review summarises the role of 3D printing in brachytherapy of gynaecological tumours.

**Abstract:**

Radiation therapy, including image-guided adaptive brachytherapy based on magnetic resonance imaging, is the standard of care in locally advanced cervical and vaginal cancer and part of the treatment in other primary and recurrent gynaecological tumours. Tumour control probability increases with dose and brachytherapy is the optimal technique to increase the dose to the target volume while maintaining dose constraints to organs at risk. The use of interstitial needles is now one of the quality indicators for cervical cancer brachytherapy and needles should optimally be used in ≥60% of patients. Commercially available applicators sometimes cannot be used because of anatomical barriers or do not allow adequate target volume coverage due to tumour size or topography. Over the last five to ten years, 3D printing has been increasingly used for manufacturing of customised applicators in brachytherapy, with gynaecological tumours being the most common indication. We present the rationale, techniques and current clinical evidence for the use of 3D-printed applicators in gynaecological brachytherapy.

## 1. Introduction

Gynaecological cancers represent an important health care burden, with cervical and endometrial cancer being the fourth and sixth most common cancers in women worldwide, respectively, and together account for more than 10% of all newly diagnosed cancers in women in 2020 [1]. Radiotherapy, including image-guided adaptive brachytherapy (IGABT) based on magnetic resonance (MR), is the standard of care in locally advanced cervical and vaginal cancer and also an integral part of curative treatment of (medically) inoperable endometrial cancer, locally recurrent cervical and endometrial cancer [2,3,4].

Tumour control probability (TCP) is influenced by tumour volume and overall treatment time [5,6,7]. TCP increases with dose and, at the same time, is higher for smaller tumours compared to large tumours at the same dose level [8,9,10]. Brachytherapy is the optimal technique to increase the dose to the target volume, while maintaining the dose to organs at risk (OARs) at set constraints [11]. In recent years, based on prospective and retrospective data collection in large groups of cervical cancer patients, new dose planning aims for MR-based IGABT with the combined intracavitary (IC) and interstitial (IS) technique were proposed [9]. Adhering to proposed dose planning aims depends on several factors, including tumour size and topography, proximity of the OARs, choice of imaging modality, choice of applicator, application technique (IC vs. IC/IS) and quality of the implant [9,12,13,14,15,16]. Ten years ago, Fokdal et al. found that 41% of patients with locally advanced cervical cancer needed interstitial needles to ensure adequate target volume coverage. Sixteen percent of all inserted needles were freehand and inserted at an oblique angle [17]. Since then, the use of interstitial needles has increased and the proportion of interstitial component use now represents one of the quality indicators for the management of cervical cancer, with at least 40% of patients treated with the combined IC/IS approach being the minimum required and ≥60% representing the optimal target [18].

Several applicators for combined intracavitary/interstitial brachytherapy have been developed in the recent decades by different companies to ensure better coverage of the target volume with the prescribed dose. However, commercial applicators to fit all anatomical variations and tumour topographies are not available. Some types of individual applicators, for example, vaginal moulds, have been used in gynaecological cancer brachytherapy for decades [19]. However, in recent years, 3D-printed applicators have been used in an increasing number of clinical situations, alone or combined with commercially available applicators to improve target volume coverage and/or overcome anatomical barriers such as a narrow vagina [20,21,22,23,24].

Three-dimensional printing, also named additive manufacturing, has been revolutionised in different fields of medicine in the last decade. First used in dentistry, its use has later spread into other medical fields, such as maxillofacial surgery, neurosurgery, urology, orthopaedic surgery, cardiology and also radiotherapy. Three-dimensional printing applications are now being used in medical training, preoperative planning and treatment [25,26,27,28,29,30,31,32]. 

In brachytherapy, 3D printing is most often used in gynaecological brachytherapy, followed by superficial brachytherapy for skin cancer and head and neck cancer [33,34,35,36,37,38,39,40]. Individual reports and small retrospective series have also been published on the use of 3D printing in lung cancer and other thoracic tumours, pancreatic cancer, liver cancer, tumours of the central nervous system, rectal cancer, breast cancer, paediatric tumours, retroperitoneal tumours and others [32,36,41,42,43,44,45,46,47]. One report suggests that 3D-printed moulds could completely replace both commercially available and hand-made moulds and become the standard of care in oral cancer brachytherapy [48].

In this paper, we summarize the rationale, techniques and current clinical evidence for the use of 3D-printed applicators in gynaecological brachytherapy.

## 2. The Rationale for Development of 3D-Printed Applicators

In a retrospective study, Petric et al. created a target density map (TDM) by merging the target volume contours of different cervical cancer patients after aligning the applicators via the centre of the ring and the ring–tandem axis, thus preserving the spatial relation of the tumour towards the applicator. Using the TDM, they assessed that the planning aims for the target can be achieved with insertion of the tandem and ring applicator in 60% of patients, while the addition of parallel needles achieved the planning aims in 95% of the tumours. For the remaining 5% of the tumours, novel applicator prototypes would need to be developed [49]. Additionally, insertion of commercially available applicators can be difficult in patients with a narrow vagina, so new solutions are also needed for some patients with smaller tumours.

The use of interstitial needles permits asymmetric modelling of isodose according to the topography of the tumour and OARs. In the past, oblique needles were inserted freehand, under transrectal ultrasound (TRUS) guidance, or via the transperineal approach. The point and angle of insertion as well as needle depth is determined in the preplanning process, based on MR images with the intracavitary applicator in place. Reproducing the preplanned needle position in a freehand insertion requires ample expertise in both needle insertion and TRUS guidance, and needle repositioning is often required. With the transperineal approach, the needle path is very long, making it difficult to keep the desired angle and direction. While this approach may be feasible for treatment of tumour extension to the lower third of the vagina, its use is unsuitable for treatment of tumour spread to the pelvic side wall or to the sacrouterine ligament.

Commercially available and modified commercial applicators such as the Vienna II, Geneva and Venezia applicators now allow the insertion of parallel and oblique needles at fixed positions and angles [50,51]. Compared to parallel needles alone, insertion of oblique needles offers a better dose distribution with fewer cold and hot spots, a lower dose to the vagina and a higher minimum dose that covers 90% of the high-risk clinical target volume (D_90_ to CTV_HR_) even for tumours extending to the distal parametria and pelvic sidewall [13,51]. However, as needle placement options are still limited in terms of both point and angle of insertion with the commercially available applicators, this could substantially impact the dose volume histogram (DVH) parameters for both the target volume and OARs, especially in large tumours, significant parametrial and/or vaginal involvement and unfavourable pelvic topography [52]. Because these cases represent a minority among all gynaecological patients treated with IGABT, and at the same time, the tumour topography in these individual patients varies from case to case, it is neither realistic nor feasible that commercial applicators will be available for these scenarios. 

The dose to the target volume is one of the most important parameters for increasing TCP, with higher doses required to achieve LC in non-squamous histological types, larger tumours and certain molecular subtypes [9,10,53]. It is therefore important to achieve planning target aims also in patients with unfavourable topography, especially in patients with large tumours and poor response to EBRT and chemotherapy. In a large, prospective, multicentre cohort study of patients with locally advanced cervical cancer, treated with curative radiotherapy including IGABT, D_90_ to CTV_HR_ and CTV_HR_ size > 45 cm^3^ were among the risk factors that had an impact on local control in multivariable analysis [10]. In a large retrospective cohort of patients with cervical cancer the use of IC/IS applicators increased D_90_ to CTV_HR_ from 83 to 92 Gy and local control in patients with a CTV_HR_ larger than 30 cm^3^ was 10% higher at 3 years with no increase in treatment-related morbidity, compared with tumours of the same size treated with IC applicators alone [54].

Achieving good implant geometry is crucial in all BT applications, as no optimisation process can correct for a poor implant. Poor implant geometry or inadequate applicator choice can compromise the dose to the target volume and OARs and negatively impact local control, acute and late toxicity [2,12]. In two prospective trials, patients whose implant was classified as inadequate had a higher risk of local failure compared to those treated with an adequate implant (HR = 2.5, *p* = 0.04). Disease-free survival (DSF) was also better in patients with an adequate implant (HR = 1.88, *p* = 0.055) [55]. The same was reported by Cornet al., who found better local control at five years in patients treated with an adequately placed applicator compared to the inadequately placed one (68%:35%, *p* = 0.02) [56]. 

The use of a customised applicator for insertion of oblique needles allows better positioning of the needles and better compliance with the preplan is usually achieved compared to freehand needle insertion [57,58]. The application is generally shorter, there is less need for needle repositioning and if using general anaesthesia, the time under anaesthesia is shorter.

Central recurrences of gynaecological tumours after surgery present another challenge, for which there are no commercially available applicators. With no uterus, the tandem cannot be used to better fix the geometry of the implant. If only the ring is used for an IC/IS application, there are too many degrees of freedom in the position of the ring within the vaginal vault, making reproducible interstitial implants very hard to achieve. At the same time, recurrences in the vagina or primary vaginal cancer that extend into the middle and/or lower third of the vagina cannot be adequately covered by such an implant. In such cases, a customised 3D-printed applicator allows a more fixed geometry, better reproducibility and better DVH parameters for target volume and OARs [14,23,59].

In postoperative radiotherapy for endometrial cancer, vaginal cylinders are used for vaginal cuff radiotherapy. The commercially available cylinders have different diameters but uniform shape and the dose is typically prescribed to a certain distance from the applicator surface. However, the postoperative size and shape of the vagina is far from uniform and there are clinical situations where the commercially available cylinders do not fit due to a narrow vagina or introitus, and air pockets form when the shape of the vaginal stump is asymmetrical, conically shaped or has a shape described as “dog ears” [24,60,61]. A 3D-printed customised mould applicator can extend the walls of the vagina, minimising the possibility of air pocket formation, ensuring better dose distribution [60].

## 3. Three-Dimensional Printing Technology

The benefit of 3D printing is the fast and relatively inexpensive production of various prototypes, which are suitable for small series and individual production. Compared to traditional manufacturing, 3D printing enables the production of more customised and complex forms. The main advantages of 3D printing technology in various areas of medicine are the reduction in production cost and time, reduction in manual work, ability to make complex geometric forms, on-demand manufacturing, personalisation and improved medical outcome [62,63]. Manufacturing could be an integral part of a BT unit or any other clinical department using 3D printing.

Medical applications produced by 3D printing include tools and medical devices, implants, medical aids and prostheses, medical models, used for educational purposes or preoperative planning, and biomanufacturing, which is a merger of 3D printing and tissue engineering [28,40,64,65,66,67,68]. In radiotherapy, 3D printing is used for the production of individual boluses in EBRT, manufacturing of (anthropomorphic) phantoms, used in medical dosimetry, equipment for the quality assurance process, training devices and the production of individual applicators and templates for IC, IS and contact BT [20,23,33,36,69,70,71,72].

There are six main methods of additive manufacturing used in medicine [62,63,73]:*Stereolythography (SLA)*—the material used is a liquid resin with photoactive mono- and polymers, which gains its final form with photopoylmerisation under UV light and high temperature. Its resolution is high, in the range of 10 μm, the surface is smooth; however, the printing is slow and expensive, and the final product is fragile.*Selective laser sintering (SLS)* or powder bed fusion (PBF)—the materials used are powders, which can be plastic, ceramic, metal or glass and are fused into solid form using a laser beam. Similar to SLA, its resolution is high, in the range of 80–250 μm, but the process is slow and costly.*Fused deposition modelling (FDM)*—the materials used are continuous fibre-reinforced polymers and filaments of thermoplastic polymers, which are heated to a semi-liquid form and ejected through the nozzle layer by layer. The method is simple, fast and cheap, with a resolution of 50–200 μm, and its major limitations are the lack of more thermoplastic materials to choose from, rough surface and mechanical fragility of the final product.*Laminated object manufacturing (LOM)*—it is used in different materials including metal, paper and polymer composites. Its advantages are low cost and a variety of materials to choose from, while its major drawbacks are poor surface quality and unsuitability for finely detailed shapes due to low dimensional accuracy.*Inkjet printing (IP)*—the material mostly used is ceramic in a form of particle dispersion, which is ejected from the printer nozzle and deposited on the surface. This method is fast, but the resolution is coarse and adhesion between layers is poor.*Direct energy deposition (DED)*—mostly metal materials in the form of powder or a wire are fused together using focused thermal energy. DED produces devices of excellent mechanical properties, and the time and costs are low; however, surface quality is poor and resolution is low at 250 μm, which renders printing of fine details hard.

The choice of the method depends on different factors such as choice of material, complexity of the product, desired resolution and cost considerations. SLS, SLA and FDM technologies are most commonly used for applicator printing in BT. The materials have to be biocompatible and certified for medical use, they have to allow some form of recurring sterilisation and hold dose attenuation properties close to water [74]. The density of most resins used for 3D printing is 1.0–1.3 g/cm^3^, so they should cause no or minor dose changes in both pulse dose rate (PDR) and high dose rate (HDR) BT [75]. The materials with high density, such as WPLA (wood polylactic acid), can be used as parts of 3D-printed individual shielded applicators [76].

Biocompatibility is a greater concern for 3D-printed applicators, used in gynaecological BT, compared to applicators used for superficial BT of the skin or 3D-printed templates for seed insertion guidance in LDR brachytherapy, as they come into contact not only with the skin but also the mucosa and blood vessels. Materials of Class VI of the U.S. Pharmacopeial Convention, Class III of the European Medicines Agency Council Directive or ISO standard 10993-certified materials should be used [50,74].

After completion, the applicator must undergo a quality control (QC) procedure, which should include mechanical QC, consisting of assessing the firmness of the applicator, testing the patency of all active channels, adequacy of fixation of different parts of the applicator and dosimetric QC. The sterilisation method should be chosen according to the type of material used for 3D printing. After sterilisation, an additional check of fixation and exclusion of possible obstruction should be performed under sterile conditions just before the insertion. QC is recommended after each sterilisation procedure. Commercially available needles and tubes should be passed through the channels and later connected to the afterloader so that the source capsule never comes into direct contact with the 3D-printed applicator.

The typical workflow for construction and use of a 3D-printed applicator is presented in Figure 1. 

Various types of applicators for gynaecological BT have been manufactured with 3D printing, including custom made vaginal cylinders that better fit the anatomy, multichannel vaginal cylinders with parallel and oblique needle channels and different add-ons for needle insertion for available commercial applicators [14,20,61,77]. Some examples of 3D-printed applicators are depicted in Figure 2.

## 4. Clinical Evidence

Most of the evidence supporting the use of 3D-printed applicators is of a low level in the form of individual case reports or retrospective series. The first report of the use of a 3D-printed applicator for cervical cancer IGABT is from Lindegaard et al., who presented the clinical workflow for the design and use of a 3D-printed vaginal template in their department. An in-house 3D printer was used and there was no delay in the treatment [21].

Wiebe et al. reported a single case of a patient with endometrial cancer, treated with BT after surgery. Due to the characteristic “dog ears” shape of the vaginal stump and a narrow introitus, a 3D-printed multichannel vaginal cylinder (MVC) in two parts was used. The two parts were assembled after insertion into the vagina. Compared to the standard single-channel cylinder, higher target volume covered by the 100% isodose (V_100_), higher D_90_ and higher minimum dose that covers 98% of the target volume (D_98_) to the CTV_HR_ were achieved, resulting in 13.2% better target volume coverage and a reduction in the target volume covered by the 200% isodose (V_200_) from 10.5 to 3.7% [24].

Sekii et al. reported two cases of patients with vaginal tumours, treated with 3D-printed templates, based on CT and MR images with a vaginal cylinder in place, presenting the workflow and reporting DVH parameters. The 3D printing was outsourced, and STL technology was used [23]. Another report on two patients with recurrent gynaecological cancer is by Laan et al., who also reported on the workflow and modelling of the applicator but did not provide dosimetric data. Sethi et al. reported on three patients with different gynaecological tumours, treated with 3D-printed vaginal cylinders due to unfavourable anatomy of the vagina, with applicator design based on gynaecological exam alone. They reported favourable DVH parameters for target volume and OARs [61].

Kang et al. published a retrospective analysis of 28 patients with gynaecological tumours, treated with low dose rate (LDR) BT. They compared 12 patients treated with 3D-printed templates for seed implantation with a group of 16 patients treated with their traditional freehand technique under CT guidance. They showed that the reproducibility of preplanned seed geometry and DVH parameters achieved with 3D-printed template guidance is better compared to freehand seed insertion [78].

Marar et al. reported on two retrospective cohorts of patients with cervical cancer, treated with 3D-printed add-ons for parallel and oblique needle guidance (TARGIT and TARGIT-FX) compared with a commercial applicator [52,79]. In the first group, they compared 302 applications in 70 patients, of which 23% were performed with the TARGIT and 77% without it, using no needles or freehand needles. V_100_, D_90_ and D_98_ for high-risk CTV (CTV_HR_) were higher in the TARGIT group, with V_100_ being higher regardless of the tumour size. There was no significant increase in doses to the OARs. The application time in the TARGIT group was longer, which could mean that the assembly of the add-on and the applicator was complicated [79]. In the second group, they compared the next-generation add-on TARGIT-FX with the original TARGIT in 148 applications performed in 41 patients. In the TARGIT-FX application, higher mean V_100_, D_90_ and D_98_ for CTV_HR_ were achieved, compared with TARGIT. The time of insertion was 30% shorter in the TARGIT-FX group. It is noteworthy that these add-ons were not individually designed for a single patient; instead, three sets of add-ons with different channel positions were designed to allow precise needle insertion through a wide range of tumour topographies [52].

Kudla et al. compared the treatment plans of ten patients with primary or recurrent tumours of the vagina, treated with a vaginal cylinder and interstitial needles inserted via a perineal template, with the theoretical treatment plans of the same patients with a 3D-printed custom-made vaginal cylinder template. The planned needle path in the tissue was shorter with the vaginal cylinder template, while the DVH parameters for the target volume and OARs were comparable or better. An interesting point is the design of needle fixation, which allows each needle to be locked individually, providing more possibilities for the needle entry point into the cylinder, compared to a mechanism which locks all the needles simultaneously [80].

In a small prospective series of nine patients with gynaecological cancer by Logar et al., all DVH parameters for both GTV and CTV_HR_ (V_100_, D_98_, D_90_ and D_100_) were significantly increased with the use of 3D-printed applicators, while the dose constraints for the OARs were not exceeded. Different applicators were used depending on the location of the tumour and patient’s anatomy—3D-printed add-on for the ring for parallel and oblique needle insertion, multichannel vaginal cylinder with channels for parallel and oblique needles, intrauterine tandem with channels for oblique needle insertion through the stopper of the tandem and a 3D-printed tandem and ring with channels for parallel and oblique needles for a patient with a narrow vagina. SLS technology was used. The advantage of using a 3D-printed applicator was shown to be independent of the size of the tumour, with sizes of CTV_HR_ ranging from 5.2 cm^3^ in a patient with local recurrence of cervical cancer after hysterectomy to 96.7 cm^3^ in a patient with primary cervical cancer [20]. 

Serban et al. published a prospective series of 20 patients with cervical cancer, treated with MR-based IGABT, including oblique needles inserted through an in-house 3D-printed vaginal template, used as an add-on for the standard tandem applicator. Additional freehand needles were inserted as needed. With a mean of 11 oblique needles per patient, excellent target volume coverage was possible with a median D_90_ for CTV_HR_ of 93 Gy even in large tumours and unfavourable topography. They also analysed the loading patterns in different parts of the applicator and reported that almost half (44%) of the dwell time was shifted to the interstitial needles, with tandem and ring dwell times accounting for 31% and 25% of the total dwell time, respectively. In this way, the dose was moved into the tumour, while the dose to the unaffected vagina was reduced and the total TRAK (total reference air kerma) remained roughly unchanged [81].

In a larger prospective study by Jiang et al., 32 patients with central recurrences of gynaecological tumours were treated with HDR BT using 3D-printed individual templates for needle insertion. Two types of applicators were printed, a transvaginal applicator with oblique channels for needle insertion for patients with vaginal stump recurrences and a combined transvaginal/transperineal applicator for patients with more extensive recurrences. There was good reproducibility of the preplanned needle positions and depth and the technique was found to be reliable and feasible [82].

Yan et al. reported an analysis of 48 patients with endometrial cancer treated with postoperative BT. They compared dosimetry of a commercial multichannel cylinder (MCC) application with 3D-printed individual MCC, modelled on CT images with a contrast-soaked vaginal packing in place. Five typical shapes of the post-hysterectomy vaginal stump were identified, and there were fewer air gaps with the 3D-printed MCC insertion. In addition, the 3D-printed MCC enabled coverage of larger CTVs with a more homogeneous dose distribution and a higher D_98_ for the CTV [60].

In the only randomised trial by Yuan et al., 21 patients with recurrent cervical cancer after surgery were randomised at the time of BT to the freehand implantation group (10 patients) and the 3D-printed guidance template group (11 patients). The D_90_ in the template group was significantly higher than in the freehand group (6.3:6.07, *p* < 0.05), while the dose to the maximally exposed 2 cm^3^ (D_2cc_) of the bladder, rectum, sigmoid and bowel was significantly lower. With a freehand implant, more needles were used (5.71:7.78, *p* < 0.05) and the procedure time was longer [14].

The studies are summarised in Table 1.

## 5. Discussion

Three-dimensional printing has gained importance over the past decade, with gynaecological tumours being the most common indication for its use in BT. The use of 3D-printed applicators represents a significant improvement in gynaecological BT, allowing implantation in patients where commercially available applicators either could not be inserted due to anatomical barriers or were not adequate for target coverage.

In cervical cancer, several studies have shown the impact of dose on TCP. A 12% increase in dose (from 75 to 85 Gy) improves local control by 3% for tumours of 20–30 cm^3^ and up to 7% for larger tumours of 70 cm^3^. An additional dose escalation from 85 Gy to 90–95 Gy can further improve local control by 1–4%, depending on tumour size [8]. In the study by Logar et al. [20], all reported dose parameters for CTV_HR_ and GTV were 30–40% better when using a 3D-printed applicator compared to the standard applicator, which could lead to a 15% better local control for stages II–III/IV taking into account the TCP curves for cervical cancer [8,20].

However, nearly 70% of studies on use of 3D printing in radiation oncology report at least one impediment or concern to the wider use of 3D printing, with the most common concerns being about time and workflow, 3D printer accuracy, biocompatibility and sterilisation of the applicator [71]. There are also some specific limitations to the use of customised 3D-printed applicators in gynaecological BT.

Due to the anatomy of the vagina, insertion of needles at a large angle through a ring-like applicator may prove difficult or even impossible due to a lack of space. Insertion into tumour infiltrates far from the ring surface (vaginal template), e.g., infiltration of the distal part of the sacrouterine ligament or the fallopian tubes, is also demanding as the trajectory of the needle can change far away from the channel exit point. The trajectories of the needle channels in an MVC are also not without limitations. Sharp angled trajectories can cause obstruction of the source wire. The attachment of 3D-printed add-ons to parts of commercial applicators can also be a challenge.

The materials used for 3D printing have mostly not been tested for repeated sterilisation, so there is limited knowledge about possible changes in the structure of the material, its firmness and potential impact on dosimetric properties. The possibilities for additional QA after sterilisation are limited, because it has to be conducted under sterile conditions immediately before implantation. For FDM, for example, several biocompatible materials are available, but only a few are reported as being able to endure the sterilisation in an autoclave, with ethylene oxide or gas plasma as recommended by the Centre for Disease Control [88]. In most reports, the 3D-printed applicators have only been used for a single patient [14,20,21,52,82], reducing the number of sterilisation procedures but increasing the cost of the application itself. There are no guidelines for the commissioning 3D-printed applicators or recommendations for the QA/QC process and widely varying levels of QC were described in the literature [20,38,74,89,90,91].

Additionally, the modelling of most 3D-printed applicators is based on a preplanning procedure, with MR or CT performed with a standard applicator in place. Based on the contours, virtual needles are placed and the applicator is modelled to accommodate the required needle trajectories. For the patient, this means an additional procedure under local, regional or general anaesthesia and an additional day of hospital stay. The cost of the additional imaging also must be taken into account.

Insertion of a large number of needles and deep insertion into the distal parametria increases the possibility of bleeding on removal of the applicator. Mahantshetty et al. reported a 27.5% incidence of bleeding at Vienna II applicator removal, with almost one-third of the bleeds being arterial [51]. This could be partly avoided by using blunt needles, provided by the vendors, and TRUS guidance for needle insertion with colour Doppler for better visualisation of blood vessels. The proximity of the blood vessels to the planned needle path should also be assessed in the preplanning MR.

There is concern that the additional steps required to use a 3D-printed applicator would prolong the overall treatment time (OTT), which could compromise local control [92]. For cervical cancer, the optimal treatment time is ≤50 days, and a dose escalation of 5 Gy is necessary to compensate for a treatment extension of 1 week [8]. Maintaining OTT at or below 50 days could be compromised by additional application and imaging required prior to 3D printing. With a 3D printer available in the department, the printing of an applicator can be completed in several hours or overnight [21,52]. Even in departments where printing was outsourced, the treatment time did not increase, but good logistics are required [20,23].

The only alternative to 3D-printed applicators, when commercially available applicators do not allow an adequate dose to the target volume, is freehand needle insertion. Studies in BT of gynaecological, head and neck and skin cancers have shown that a template-guided implant achieves better DVH parameters, better reproducibility of the preplanned needle positions and better adaptation to patient anatomy than freehand insertion [14,33,35,36,57,93]. Huang et al. reported on the accuracy of template-guided needle insertion in 25 patients with head and neck cancer [36]. All 619 interstitial needles were inserted to the planned position on the first try, the mean deviation from the entry point in the preplan was 1.18 mm and the mean angular displacement was 2.08°. The only randomised trial comparing 3D-printed template-guided with freehand insertion of oblique needles in gynaecological tumours also confirmed better needle positioning with significantly better DVH parameters for both the target and for the OARs [14]. In addition, with template-guided insertion, there is less need for potential needle repositioning during the procedure and after a post-implant MR [78]. If the MR shows the need for additional needles, needle repositioning or depth correction, additional operating room time and post-correction imaging will be required, all of which increase the costs and use of departmental resources. The use of TRUS guidance can reduce the need for needle repositioning after post-implant MR; however, it is sometimes difficult even for a skilled radiation oncologist to assess the adequacy of needle position within the target with TRUS or abdominal US.

The number of oblique and parallel needles that can be used for the manufacturing of individual applicators is limited and geometrically determined by the trajectories of the needle channels which must not intersect or merge. To our knowledge, there is currently no software on the market that can be used as a tool in the preplanning process to determine the optimal positions of the needles within the target volume to achieve the best coverage of the target volume with the minimum number of needles. In this way, a virtual optimal distribution of needles within the individual applicator would be achieved, which would speed up not only the preplanning process but also the application procedure itself. An experienced multidisciplinary team should be involved in the process. If modelling and 3D printing are not outsourced, additional staff training might be needed. The strengths and limitations of 3D-printed applicator use are summarised in Table 2.

## 6. Conclusions and Future Directions

Three-dimensional printing is a promising and still developing technique in gynaecological BT. The use of customised applicators is necessary in a minority of patients with gynaecological cancers and should be performed in large-volume BT centres with experienced radiation oncologists and physicists. While customised applicators are economically unattractive for large manufacturers of radiotherapy equipment due to the small number of cases, smaller companies specialising in 3D printing of various medical equipment could emerge. As 3D printer prices decrease and 3D printing materials become cheaper, the wider use of this technology in clinical departments can be expected. The radiation oncology community should form focus groups to develop guidelines for the manufacture and commissioning of 3D-printed applicators, with emphasis on the QA/QC process, which currently varies widely among centres already using 3D printers.

In the future, using both low and high Z materials for 3D printing, shielded applicators enabling intensity-modulated BT and protection of OARs could be produced. Some dosimetry reports and phantom studies have already been published, but clinical data are lacking [39,91,94,95]. Larger prospective studies on the efficacy and safety of 3D-printed applicators are also needed before 3D printers come to be a part of our daily clinical practice in a BT department.

Another recent revolution in medicine is the introduction of the Internet of Things (IoT) concept in various healthcare settings [96,97,98,99]. It is most widely used in neurology and cardiology [98], but there are some reports of its applicability in oncology and radiation oncology [100,101,102,103,104]. Virtual reality, artificial intelligence (AI) and robotics are components of the IoT, for which use in BT has already been described [103,104]. For the emerging field of 3D printing in brachytherapy, the IoT presents an interesting opportunity to remotely connect machines and experts in the field to enable access to cutting-edge treatment even in BT centres where the technology or knowledge is not available. In surgery, the possibility of tele-surgery and tele-mentoring is already being explored [99]. Similarly, an experienced radiation oncologist and physicist could remotely perform or assist with preplanning, 3D modelling and applicator insertion. AI could help with the preplanning process, suggesting the optimal needle trajectories to improve coverage of the target volume and perform 3D modelling of the applicator based on large data sets available in the IoT, which would speed up the process. However, there are still many challenges to overcome, especially in the areas of data monitoring, governance and ownership but also in reimbursement, and studies are needed to test the clinical relevance of IoT in gynaecological BT.

## Figures and Tables

**Figure 1 cancers-15-04165-f001:**
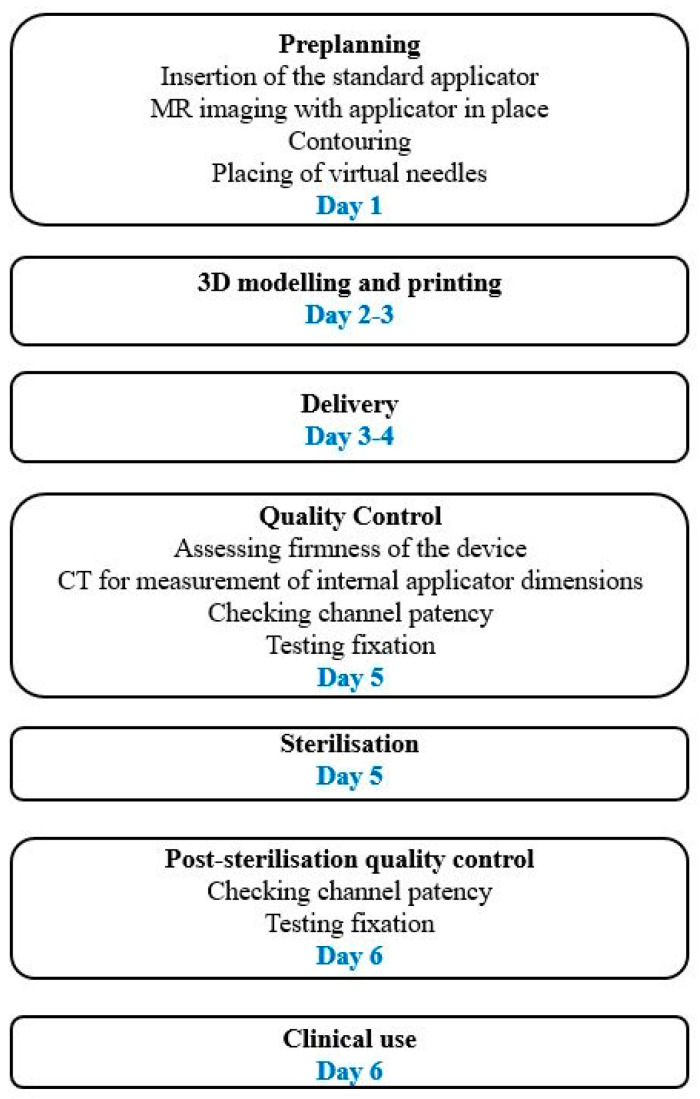
Typical workflow for the design and use of a 3D-printed applicator when outsourcing the printing. Workflow can be shortened when an in-house printer is used.

**Figure 2 cancers-15-04165-f002:**
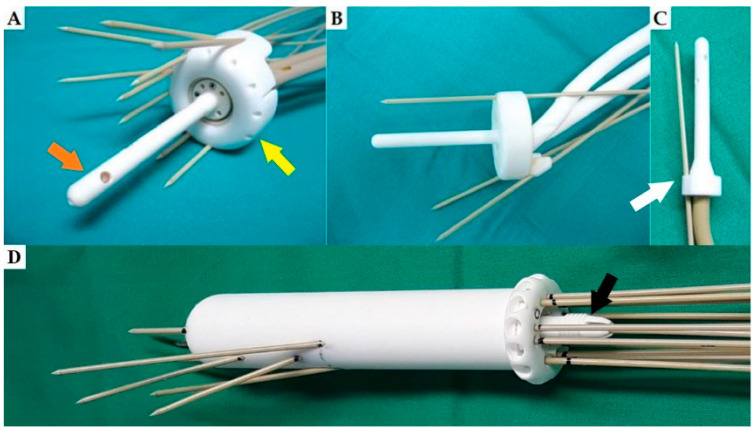
Examples of 3D-printed applicators used in our department. (**A**) The 3D-printed intrauterine tandem (orange arrow) and add-on for parallel and oblique needle insertion for the ring (yellow arrow). (**B**) The 3D-printed tandem and ring with channels for parallel and oblique needles. (**C**) The 3D-printed intrauterine tandem with oblique needle channels in the stopper (white arrow). (**D**) The 3D-printed vaginal cylinder with parallel and oblique needles. Needle fixation screw is marked with a black arrow. Needles provided by the vendor were used in all cases.

**Table 1 cancers-15-04165-t001:** Summary of the studies on the use of 3D-printed applicators in gynaecological brachytherapy. Only the studies with clinical cases are included.

Author, Publication Year	Type of Study/No. of Pts	Patient Selection	Type of 3D-Printed Applicator	Results
Yuan et al., 2019 [14]	Randomised/21 pts	Recurrent cervical cancer	VC with oblique needles, compared to freehand	Higher D_90_ for CTV_HR_, lower D_2cc_ all OARs; Fewer needles with 3D cylinder
Yan et al., 2021 [60]	Prospective/48 pts	Postoperative endometrial cancer	MVC, compared to commercial cylinder	Higher D90 for CTV, more homogeneous dose, fewer air pockets
Jiang et al., 2020 [83]	Prospective/32 pts	Central recurrences	MVC with oblique needles	Good reproducibility of preplanned needle positions, technique feasible
Logar et al., 2019 [20]	Prospective/9 pts	Primary and recurrent gyn tumours	Depending on tumour type (see text), compared to standard applicator	V_100_, D_98_, D_90_ and D_100_ for GTV and CTV_HR_ higher compared to standard applicator
Kudla et al., 2023 [80]	Retrospective/10 pts	Primary and recurrent vaginal tumours	MVC with oblique needles, compared to transperineal implant	Shorter needle path with 3D applicator; Similar DVH parameters
Marar et al., 2022 [79]	Retrospective/70 pts	Cervical cancer	TARGIT add-on for T&O, compared to T&O	V_100_, D_90_, D_98_ for CTV_HR_ higher with TARGIT, longer insertion time
Marar et al., 2023 [52]	Retrorospective/41 pts	Cervical cancer	TARGIT FX add-on for T&O, compared to TARGIT	V_100_, D_90_, D_98_ for CTV_HR_ higher with TARGIT-FX Insertion time 30% shorter
Kang et al., 2021 [78]	Retrospective/28 pts	Gynaecological tumours	Template for seed insertion, compared to freehand	Better reproducibility of preplanned seed geometry
Sekii et al., 2018 [23]	Case report/2 pts	Vaginal tumours	MVC with oblique needles	Presenting workflow, reporting DVH parameters
Sethi et al., 2016 [61]	Case report/3 pts	Gynaecological tumours	Customised MVC	Favourable DVH parameters for target and OARs
Laan et al., 2019 [77]	Case report/2 pts	Recurrent gyn tumours	Personalised needle template	Presenting workflow and applicator modelling No DVH data
Lindegaard et al., 2016 [21]	Case report/1 pt	Cervical cancer	Tandem and 3D-printed ring-like template	Presenting workflow, applicator modelling and DVH parameters
Wiebe et al., 2015 [24]	Case report/1 pt	Postoperative endometrial cancer	Customised MVC, compared to standard single-chanel VC	V_100_, D_90_, D_98_ for CTV higher, V_200_ lower 13.2% better coverage
Sohn et al., 2022 [75]	Retrospective/5 pts	Cervical cancer	3D vaginal template + T&O, compared to T&O + freehand needles	Better optimality, target coverage and OAR sparing
Qin et al., 2022 [84]	Prospective/9 pts	Recurrent cervical cancer	MVC with needles, compared to commercial single-channel VC	Planning aims achieved in all 3D print plans, but failed in 3 VC plans
Serban et al., 2021 [81]	Retrospective/20 pts	Cervical cancer	3D vaginal template + T&R + freehand needles	CTV_HR_ D90 93 Gy, D_2cc_ bladder/rectum/sigmoid/bowel 78/65/59/61 Gy
Liao et al., 2022 [85]	Prospective/6 pts	Postoperative endometrial cancer	Template for VC fixation	Better reproducibility of VC position, less difference in D_2cc_ btw fractions, non-significant
Zhang et al., 2019 [86]	Case report/3 pts	Cervical cancer	Customised IC/IS applicator, compared to standard applicator, inverse planning	Higher D_90_ for CTV_HR_, better OAR sparing compared to standard applicator
Liu et al., 2021 [87]	Retrospective/103 pts	Recurrent cervical cancer post-EBRT	Template for non-coplanar ^125^I seeds implantation	Safe, effective, minimally invasive, 1 > G2 acute, 2 > G2 late adverse events

No. = number, pts = patients, OARs = organs at risk, T&O = tandem and ovoids, T&R = tandem and ring, VC = vaginal cylinder, MVC = multichannel vaginal cylinder, D_2cc_ = dose to maximally irradiated 2 cm^3^ of OAR, gyn = gynaecological, EBRT = external beam radiotherapy, btw = between.

**Table 2 cancers-15-04165-t002:** The strengths and limitations of 3D-printed applicators in gynaecological brachytherapy.

Strengths	Limitations
Fast production of customised and complex forms	Additional steps needed (application, imaging, preplanning, modelling, QA/QC)
Allow complex geometry, oblique angles, non-coplanar needle distribution	New skills required, additional education
Shorter applications—less time in OR	Accuracy of 3D printers
Better position accuracy, favourable geometry	Materials not tested for repeated sterilisation
Better reproducibility, consistent placement	Limited possibility for post-sterilisation QA/QC
Higher dose to target volume—better local control	Material biocompatibility issues
Reducing dose to OARs	No guidelines for applicator manufacture, commissioning and QA/QC
Reducing patient discomfort	Applicator fixation issues
Better fit to patient’s anatomy	Potentially prolonged OTT
Possibility of shielded applicators	Potentially increased costs
	Lack of good quality prospective clinical data

OR = operating room, QA/QC = quality assurance/quality control, OTT = overall treatment time, OARs = organs at risk.

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
