# Peer review of "The Use of 3D Printing Technology in Gynaecological Brachytherapy—A Narrative Review"

_cancers, 2023, doi:10.3390/cancers15164165_

Round 1
Reviewer 1 Report
Interesting and current work
There are typographical errors.
I suggest also the reference "Individual 3-dimensional printed mold for treating hard palate carcinoma with brachytherapy: A clinical report. Journal of Prosthetic Dentistry 2019;121(4), pp. 690-69322 The paper suggests that individual 3D printed molds have the potential to replace commercially available materials with handmade molds, their use may well become standard for a lot of treatment with brachytherapy.
The authors should better define the strengths and limitations of 3D 3D printing
Author Response
Dear reviewer,
Thank you for your feedback and critical remarks.
The text has been thoroughly read through to identify typographical errors, additional modifications were done to further improve the language and spelling.
The suggested reference has been added.
To better emphasise the strengths and limitations of 3D printing we have decided to add Table 2. in the discussion section.

Reviewer 2 Report
Informative and comprehensive review detailing techniques along with the clinical evidence in using 3D printed applicators in gynaecological brachytherapy.
A good use of relevant literature to critically evaluate and formulate the rationale for 3D printed applicator technology to treat a range of gynaecological diseases.
Feedback would be to annotate (arrows or other numerical labelling) the specific components in figure 2 and also in figure one adding a timeframe to put the workflow into a real life context.
Author Response
Dear reviewer,
Thank you for your feedback and critical remarks.
In Figure 2, arrows have been added to facilitate the understanding of different parts of the applicators, different applicators have already been marked from A to D and addressed accordingly in the captions. In Figure 1, a time frame in days has been added, the example of our department was used, where 3D printing is outsourced. The possibility to shorten the time frame with an in-house printing device was addressed in the captions, as well as the text.

Reviewer 3 Report
I was glad to review the work of the authors regarding this very interesting review on the use of 3D printing technology in gynecological brachytherapy. The manuscript is well-written and the incorporated table and figures make the study easy to follow.
I strongly recommend acceptance for publication of the paper after minor changes.
This review summarizes the rationale, techniques, and current clinical evidence for the use of 3D-printed applicators in gynecological brachytherapy.
“In the last few years, technological developments in the medical/surgical field have been rapid and are continuously evolving. One of the most revolutionizing breakthroughs was the introduction of the IoT concept within the medical/ surgical practice.”
Add this information in the discussion section and explain the role of IoT in the use of 3D-printed applicators.
Consider citing the article on the Internet of surgical things
https://pubmed.ncbi.nlm.nih.gov/35746359/
General comments
The spelling and punctuation are very good. No issues were detected.
Abstract
The abstract is concise. All the necessary information about the study is included.
Background
- The information provided in the introduction is important for the comprehension of the article.
- The objective of the study is clearly mentioned.
Methods
- The methods are sufficiently explained by the authors.
Results
- The results are presented in a very extensive way.
- The table is really helpful and necessary for the completion of the authors' work.
Discussion
- The discussion is of great quality and includes updated data.
- The authors inform the reader about the study's limitations.
Conclusion
From the presented data, the conclusion is complete and represents the work that the authors did.
Author Response
Dear reviewer,
Thank you for your feedback and critical remarks.
The topic of IoT has been added into the conclusions and future directions section. The suggested paper on the internet of surgical things has been cited, along with some other papers on the same subject.
